# Quantitative 3D-imaging for cell biology and ecology of environmental microbial eukaryotes

**Sebastien Colin[1,2,3†]\*, Luis Pedro Coelho[4†], Shinichi Sunagawa[4‡], Chris Bowler[5], Eric Karsenti[5,6], Peer Bork[4], Rainer Pepperkok[3,7], Colomban de Vargas[1,2]\***

[1]UMR 7144, team EPEP, Station Biologique de Roscoff, Centre Nationnal de la Recherche Scientifique, Roscoff, France; [2]Université Pierre et Marie Curie, Sorbonne Universités, Roscoff, France; [3]Advanced Light Microscopy Facility, European Molecular Biology Laboratory, Heidelberg, Germany; [4]Structural and Computational Biology, European Molecular Biology Laboratory, Heidelberg, Germany; [5]Institut de Biologie de l'École Normale Supérieure, École Normale Supérieure, Paris Sciences et Lettres Research University, Paris, France; [6]Directors' Research, European Molecular Biology Laboratory, Heidelberg, Germany; [7]Cell Biology and Biophysics Unit, European Molecular Biology Laboratory, Heidelberg, Germany

**Abstract** We present a 3D-fluorescence imaging and classification tool for high throughput analysis of microbial eukaryotes in environmental samples. It entails high-content feature extraction that permits accurate automated taxonomic classification and quantitative data about organism ultrastructures and interactions. Using plankton samples from the *Tara* Oceans expeditions, we validate its applicability to taxonomic profiling and ecosystem analyses, and discuss its potential for future integration of eukaryotic cell biology into evolutionary and ecological studies.
DOI: https://doi.org/10.7554/eLife.26066.001

**\*For correspondence:**
colin@sb-roscoff.fr (SC); c2vargas@gmail.com (CV)

†These authors contributed equally to this work

**Present address:** ‡Department of Biology, Institute of Microbiology, ETH Zurich, Zurich, Switzerland

**Competing interests:** The authors declare that no competing interests exist.

## Introduction

Studies of organismal diversity have traditionally focused on plants and animals, and more recently on prokaryotes and viruses, while much less attention has been given to microbial eukaryotes (*Pawlowski et al., 2012*; *del Campo et al., 2014*) (mostly unicellular protists, plus small multicellular organisms between 0.5 µm and 1 mm in size). On the other hand, genetics together with molecular, cell, and developmental biology have used a limited number of model organisms to reveal over the past 40 years key principles underpinning the organization of living matter (*Alberts, 2014*). However, recent holistic meta-omics surveys of biodiversity across the full spectrum of life (*Bork et al., 2015*) are revealing nowadays a massive amount of unknown taxa and genes in microbial eukaryotes (*de Vargas et al., 2015*) implying that much more eukaryotic diversity and functions need to be explored. Phylogenomics (*Burki et al., 2016*), geological (*Knoll, 2014*; *Falkowski and Knoll, 2007*), and ecological (*de Vargas et al., 2015*; *Mahe et al., 2016*) records also show how intense symbiogenesis and diversification in protists have shaped the evolution of eukaryotes together with that of biogeochemical cycles. In order to better understand how such processes have led to the complexification of life and the Earth system, it is necessary to bridge contemporary diversity studies to environmental eukaryotic cell biology in an ecological context. As a first step, we need new automated high-content 3D-imaging systems that can cope with the broad size, abundance, and complexity ranges characterizing environmental microbial eukaryotes.

Automated imaging techniques to tackle the diversity of uncultivated aquatic organisms include in-flow systems that couple high-throughput low-resolution imaging with feature extraction from

**eLife digest** Our planet's ecosystems – from its oceans to its forests – are teeming with microbes. DNA analysis of environmental samples shows that many of these microbes belong to a group known as protists. This group consists of single-celled organisms that are close relatives of fungi, plants and animals. Though protists are a widespread and diverse group, scientists know little about them. One reason for this is the lack of high-throughput ways to recognize and count protists in environmental samples.

Colin, Coelho et al. set out to tackle this blind spot in ecology and cell biology by developing an automated imaging system. The system needed to image many kinds of protist cells in enough detail to see the features inside. The end-result was a 3D-imaging technique called e-HCFM – which is short for "environmental high content fluorescence microscopy". Colin, Coelho et al. went on to use the technique on 72 samples collected on an expedition across the world's oceans. This allowed them to automatically image, recognize and classify over 330,000 organisms.

This approach and new dataset will benefit researchers working in many fields, from cell biology to ecology, computational biology and beyond. In the future, this imaging method might integrate with techniques that can analyze the DNA in individual cells. This would allow scientists to link protists' visible features to their genetic information, in a way that will scale from single cells up to entire ecosystems.

DOI: https://doi.org/10.7554/eLife.26066.002

single micro-organisms (*Sieracki et al., 1998*; *Sosik and Olson, 2007*), and widefield microscope-based methods applied to phytoplankton recognition and quantification (*Embleton et al., 2003*; *Rodenacker et al., 2006*; *Schulze et al., 2013*). These approaches can characterize eukaryotic organisms from low contrast bright field images acquired at a single focal plane, sometimes in association with auto-fluorescence measurements of photosynthetic pigments (*Schulze et al., 2013*; *Hense et al., 2008*). The identification and classification of eukaryotes is then based on semi-automated machine learning approaches for a few to tens of taxonomic classes (*Benfield et al., 2007*) with a peak performance at 70–90% accuracy, which is comparable to what trained annotators can achieve (*MacLeod et al., 2010*; *Culverhouse, 2007*). None of these tools provide images of sufficient quality and resolution to assess the structural complexity of eukaryotic cells whose diversity peaks in the 5 to 50 μm size-range (*de Vargas et al., 2015*).

To improve both information content and the automated capture of morphological complexity of microbial eukaryotes over a broad size-range, we developed a 3D multichannel imaging workflow denoted e-HCFM (environmental High Content Fluorescence Microscopy) (*Figure 1a*). Here, we introduce this new method, which adapts recent protocols developed in model cell biology (*Pepperkok and Ellenberg, 2006*; *Eliceiri et al., 2012*) to the large diversity of scattered objects that characterizes environmental samples (organisms, debris, aggregates, abiotic objects, etc.). We then apply e-HCFM to 72 plankton samples collected during the *Tara* Oceans expeditions (*Karsenti et al., 2011*) (*Figure 1—source data 1*), containing communities of planktonic organisms in the 5–20 μm size range, which are typically analyzed automatically by flow cytometry with a very low morpho-taxonomic resolution (*Marie et al., 2014*; *Zubkov et al., 2007*). We show that e-HCFM allows its users (i) to obtain accurate 3D images of micro-eukaryotes over a broad taxonomic range; (ii) to enrich significantly the information content of each imaged organism; (iii) to segregate automatically the high diversity of planktonic micro-particles, and (iv) to archive annotated digital images of perishable samples. These advances allow linking identification and quantification of uncultured eukaryotes with ecological and functional traits.

## Results and discussion

Size-fractionated plankton communities collected across the world oceans during the *Tara* Oceans expeditions (*Pesant et al., 2015*) (*Figure 1a*) and fixed onboard were used as starting material. Prior to imaging, sample aliquots are loaded into an optical chamber coverglass slide, which is centrifuged to concentrate the organisms onto the glass-bottom of the well, coated for promoting cell adherence (*Figure 1a*, *Figure 1—figure supplement 3*). Subsequent preparation steps (rinsing/staining/

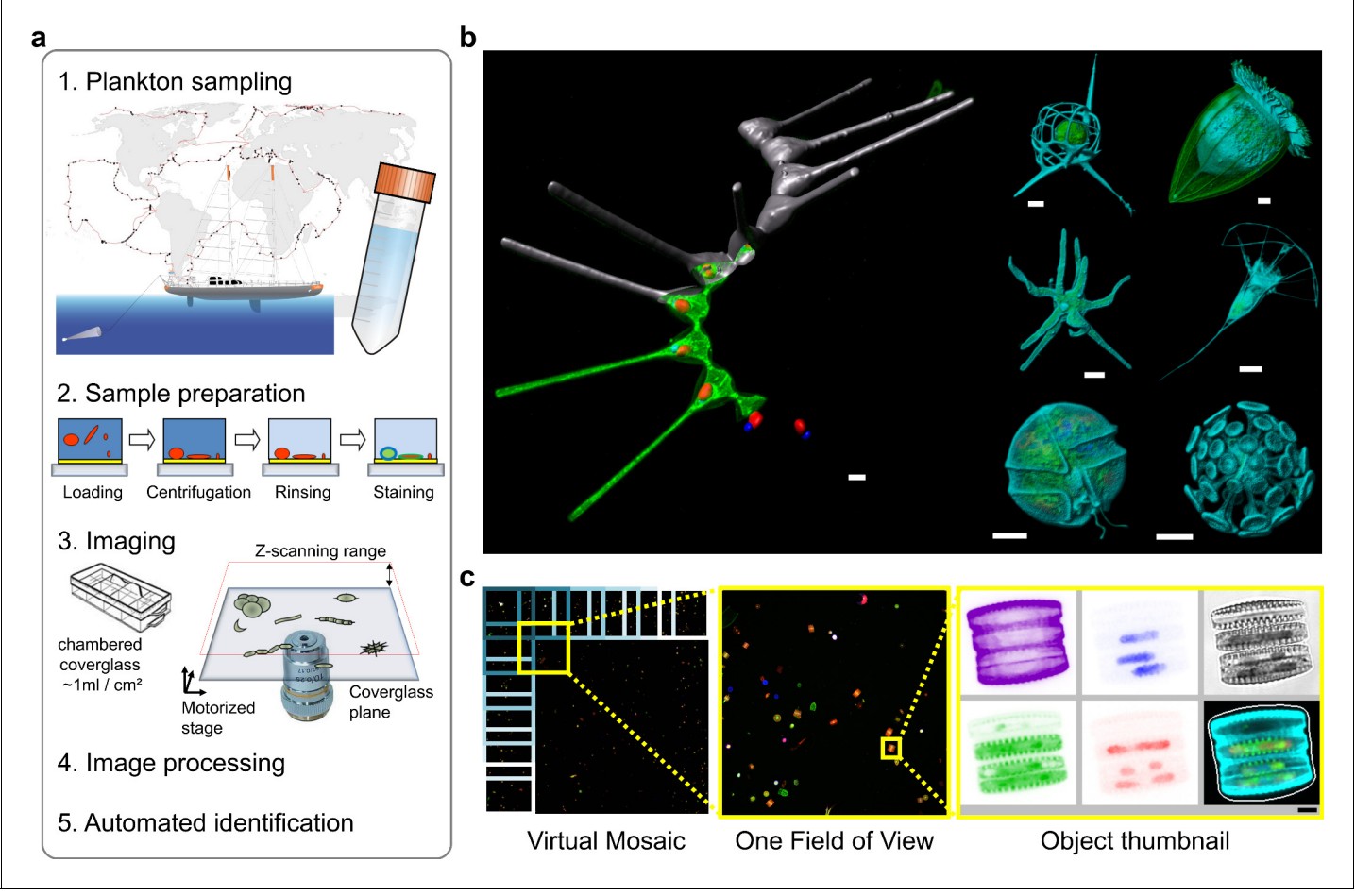

**Figure 1.** Environmental high-content fluorescence microscopy (e-HCFM): automated, 3D, and multichannel imaging for aquatic micro-eukaryotes. (**a**) e-HCFM workflow applied to *Tara* Oceans samples: (1) 72 nano-plankton (size range 5–20 µm) samples collected during the *Tara* Oceans expedition (*Pesant et al., 2015*) were fixed in paraformaldehyde-glutaraldehyde buffer onboard and kept at 4°C for up to several years; (2) Samples were mounted in optical multiwell plates. Then, a 4-steps preparation allowed to stain all eukaryotic cells; (3) A commercial confocal laser scanning microscope was used to automatically image samples (40X NA1.1 water lens; 5 channels) generating 2.5 Tb of raw data (acquisition details in *Figure 1—source data 1*); (4) In total, 336,655 objects were processed for individual extraction of 480 descriptors (3D biovolumes, intensity distribution, shape descriptors and texture features, details in *Figure 1—source data 2*), and the reconstruction of various images for visual inspection (**c**); (5) A training set based on 18,103 manually curated images (5.4% of the dataset) classified into 155 categories, was used for automated recognition (Random Forest). (**b**) Examples of e-HCFM 3D-images and movies from a wide phylogenetic diversity of planktonic eukaryotes (see also *Figure 1—figure supplements 1* and *2*). Left panel: a chain of diatoms (*Asterionellopsis* sp., Heterokonta) (*Figure 1—video 1*); right panel, top left to bottom right: radiolarian (Rhizaria), ciliate (Alveolata), amoeba (Amoebozoa), choanoflagellate (Opisthokonta), dinoflagellate (Alveolata), coccolithophore (Haptophyta). Key cellular features are labelled with various dyes: DNA/nuclei (blue, Hoechst33342); (intra)cellular membranes (green, DiOC6(3)); cell covers and extensions (cyan, PLL-AF546, a home-made conjugation between α-poly-L-lysine (PLL) and Alexa Fluor 546 (AF546)); chloroplasts (red, chlorophyll autofluorescence). Scale bar 5 µm. (**c**) The confocal microscope is automated for acquiring multicolor Z-stacks over a mosaic of positions for each sample. Each field of view (fov) overlaps neighboring ones for detection of entire cells even if their position crossed one fov edge (see also *Figure 1—figure supplement 4a*). The imaging volume along the Z-axis comprises the space between the coverglass/sample interface plane and an upper limit corresponding to the theoretical thickest cell (*Figure 1—figure supplement 3b*). The fovs are then processed automatically and sequentially for detecting organisms without redundancy and for generating various images and Z-stack animations for visual inspection (*Figure 1—videos 2* and *3*).

DOI: https://doi.org/10.7554/eLife.26066.003

The following video, source data, and figure supplements are available for figure 1:

**Source data 1.** This image acquisition registry details the e-HCFM imaging runs, their metadata, their samples of origin, and associated metadata from the Tara Oceans expedition.

DOI: https://doi.org/10.7554/eLife.26066.011

**Source data 2.** List of descriptors computed for each object imaged through e-HCFM.

DOI: https://doi.org/10.7554/eLife.26066.012

**Figure supplement 1.** e-HCFM staining strategy is adapted to the environmental protistan biodiversity.

*Figure 1 continued on next page*

*Figure 1 continued*

DOI: https://doi.org/10.7554/eLife.26066.004

**Figure supplement 2.** e-HCFM staining strategy is suitable for live imaging.

DOI: https://doi.org/10.7554/eLife.26066.005

**Figure supplement 3.** e-HTFM strategy for automated screening of planktonic protists.

DOI: https://doi.org/10.7554/eLife.26066.006

**Figure supplement 4.** The image acquisition is coordinated with the image-processing strategy.

DOI: https://doi.org/10.7554/eLife.26066.007

**Figure supplement 5.** A short selection of the biodiversity that was detected by e-HCFM into 5–20 µm samples from the Tara Oceans expeditions.

DOI: https://doi.org/10.7554/eLife.26066.008

**Figure supplement 6.** Monitoring the microscope performance over time by measuring the fluorescence intensity of three red fluorescing beads (Inspeck deep red 0.32% [top panel]; 1.41% [middle panel]; 4.10% [lower panel], I-7225, Invitrogen) in the chlorophyll channel (ex633/em680-700).

DOI: https://doi.org/10.7554/eLife.26066.009

**Figure supplement 6—source data 1.** Bead intensities for each bead.

DOI: https://doi.org/10.7554/eLife.26066.010

**Figure 1—video 1.** 3D-animation of a Diatom (*Asterionellopsis* sp.) which illustrates the synergy between e-HCFM staining strategy and optical sectioning microscopy (CSLM).

DOI: https://doi.org/10.7554/eLife.26066.013

**Figure 1—video 2.** First example of Z-stack animations that e-HCFM method provides for each detected organism (a chain of diatoms).

DOI: https://doi.org/10.7554/eLife.26066.014

**Figure 1—video 3.** Second example of Z-stack animations that e-HCFM method provides for each detected organism (a choanoflagellate).

DOI: https://doi.org/10.7554/eLife.26066.015

washing) are minimized to avoid cell loss and manipulations, and thus preserve fragile bodies for reliable quantification. Two environmental samples are prepared manually in 1 hr. Key cellular features were targeted with a combination of fluorescent dyes and auto-fluorescence (*Figure 1b*, *Figure 1—video 1*): (i) DNA content; (ii) (intra)cellular membranes and hydrophobic bodies; (iii) chloroplasts and potential photosymbionts, and (iv) the overall contour of eukaryotic cells including cell walls and external matrices. For this last purpose, we specifically designed the PLL-AF546 dye (conjugation between α-poly-L-lysine and Alexa Fluor 546, see method section below) that labels a wide range of exoskeletons made of polysaccharides, proteins, carbonate, silicate, etc., abundant in marine microbial eukaryotes (*Figure 1b* and *Figure 1—figure supplements 1* and *2*). The fluorochrome moiety can be easily adapted for other multicolor labeling protocols, and the staining strategy can be applied to either live (*Figure 1—figure supplement 2*) or fixed cells in suspension (*Figure 1—figure supplement 1*). Overall, our protocol is highly effective at revealing internal and external cellular structures in organisms representing the known phylogenetic diversity of eukaryotes (*Figure 1—figure supplement 1*), including their diverse biotic interactions (*Figure 2*, *Figure 2—video 1*, and *Figure 2—figure supplement 1*).

We used a confocal microscope (Leica Microsystem SP8) equipped for automated high-content imaging. The magnification and resolution are adjusted by selecting a suitable lens for the size-range of the organisms and structures of interest. While the XY size range (1 to 1500 µm) is limited by the fields of view (fov) of the lenses (*Figure 1c*), the range in the Z-axis (thickness) is constrained by the transparency of the object. Objects are optically sectioned to spatially resolve the fluorescence signal, resulting in sharp images of the entire cells. For each imaged position, five channels are recorded (bright field and four fluorescent signals) and the raw imaged Z-stacks are then segmented to extract single planktonic particles (*Figure 1c*, *Figure 1—figure supplement 4*). Particles are further processed individually to sub-segment regions of biological interest (e.g. nuclei or chloroplasts) for each fluorescent channel. A collection of 480 numeric 2D/3D features (*Figure 1—source data 2*) is computed for each captured particle, including biovolumes, intensity distribution, shape descriptors and texture features. The features were used in combination with a visually curated learning set (see below) to build a machine learning model (*Boland et al., 1998*), which associates each particle to a category (taxonomic or morphological) of the training set (*Figure 3a*; *Figure 3—source data 1*).

Imaging and processing of each sample with a 40x lens lasted 8 hr, leading to 396 fovs (*Figure 1a,c*), over a 22 µm Z-range. Overall, e-HCFM imaging of the 72 *Tara* Oceans plankton

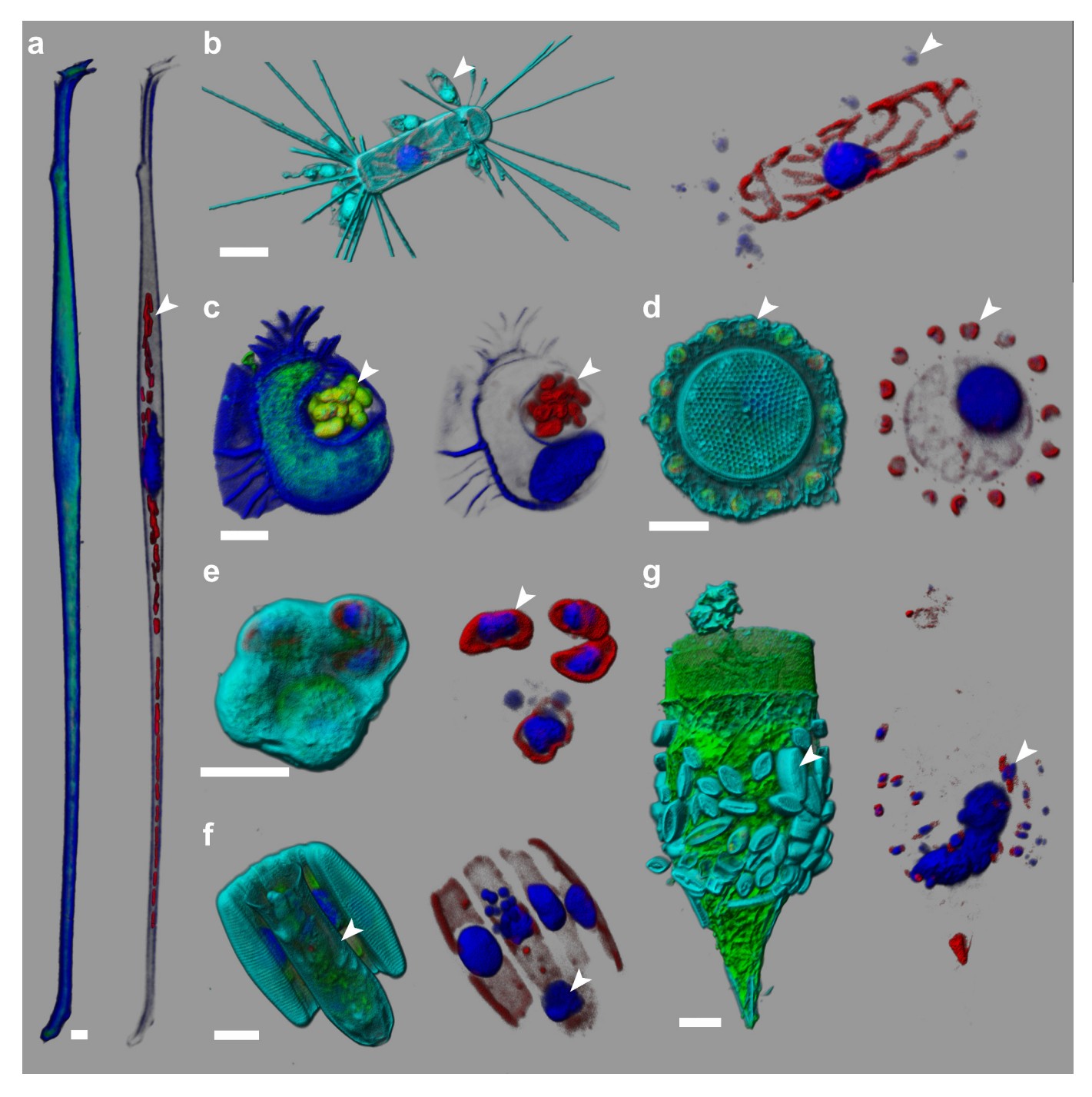

**Figure 2.** e-HCFM-staining strategy is effective in revealing symbiotic interactions in marine protists. These seven cells, fixed on board *Tara* and kept at 4°C for several years, were imaged manually using the e-HCFM workflow (*Figure 1*). Each cell is illustrated by two panels: the left side overlays all available fluorescent channels whereas the right side displays only the chlorophyll and the Hoechst fluorescence. Four fluorescent channels were recorded: (i) Green: cellular membranes (DiOC6(3)) indicate the core cell bodies; it also stains loricas of tintinnid ciliates (g); (ii) Blue: DNA (Hoechst) identifies nuclei; it also stains the cell-wall of thecate dinoflagellate (a, c); (iii) Red: chlorophyll autofluorescence resolves chloroplasts; (iv) Cyan: PLL-A546 is a generic counterstain for visualizing eukaryotic cells' surface (not used in a, (c). 3D reconstructions were conducted with the software Imaris (Bitplane). Scale bar is 10 µm. (a) Association between the heterotrophic dinoflagellate *Amphisolenia* and unidentified cyanobacteria hosted inside the cell wall (arrow head). (b) The diatom *Corethron* sp. (*Figure 2—video 1*) harbors several epiphytic nanoflagellates living in small lorica and attached onto the diatom frustule (arrow head). These have been observed in association with different diatom species (see *Figure 2—figure supplement 1*). (c) The dinoflagellate *Citharistes* sp. has developed a chamber (phaeosome) for housing cyanobacteria (arrow head). (d) The diatom *Thalassiosira* sp. is

*Figure 2 continued on next page*

*Figure 2 continued*
surrounded by a belt chain of 14 coccolithophores (*Reticulofenestra sessilis*, arrow head). (**e**) A juvenile pelagic foraminifer hosts endosymbiotic microalgae (arrowhead), likely *Pelagodinium* dinoflagellates. (**f**) Colonies of *Fragillariopsis* sp. diatoms are regularly observed in close interaction with tintinnid *Salpingella* sp. ciliate (arrowhead). The tintinnid lorica is inserted inside the groove of the barrel formed by the diatom chain. (**g**) The lorica of the ciliate *Tintinnopsis* sp. aggregates several epiphyte pennate diatoms, which were still alive prior to fixation as chloroplast and nuclei are visible (arrow head).
DOI: https://doi.org/10.7554/eLife.26066.016
The following video and figure supplement are available for figure 2:

**Figure supplement 1.** e-HCFM reveals an unreported epibiose involving the diatom *Chaetoceros simplex* and an unidentified nano-flagellate from the 5–20 µm samples of the Tara Oceans expeditions.
DOI: https://doi.org/10.7554/eLife.26066.017
**Figure 2—video 1.** 3D-animation of a Diatom (*Corethron* sp.) which illustrates how the e-HCFM method supports investigation about microbial interactions.
DOI: https://doi.org/10.7554/eLife.26066.018

samples (*Figure 1—source data 1*) produced ~2.5 Tb of raw data, corresponding to 336,655 planktonic particles, each of them associated with a series of derived images for visual inspection including summary thumbnails (*Figure 1c*, *Figure 1—figure supplements 4* and *5*), Z-stack animations, 3D segmentation masks, and 3D reconstructions. The segmentation performance of each particle is enhanced by the sharp contrast generated by the fluorescent signals. In particular, the thin transparent biomineral and/or organic structures shaping many groups of eukaryotic taxa can be precisely detected (*Figure 1—figure supplement 5*), generating more accurate computed features.

We then generated a reference training set to build an automated classifier. A total of 18,103 objects (5.4% of the whole dataset) were manually curated and classified into a 4-level hierarchical framework including 155 categories (taxonomic lineage for organisms and morphological types for other particles, *Figure 3a*; *Figure 3—source data 1*). A random forest classifier was empirically determined to perform the best, combining high accuracy and fast computation time. We estimate the overall accuracy of the classifier at 82.2% at the finest taxo-morphological level (corresponding often to genus level), rising to 93.8% at the phylum/class level (*Figure 3a*, *Figure 3—source datas 1* and *2* and *Figure 3—figure supplement 1*). This shows that our cell descriptors provide efficient segregation of the main plankton taxonomic groups, despite considerable intra-group morphological diversity. The same method applied on image descriptors obtained only from the bright field channel (which are a subset of the 480 features used in this study) resulted in only 56.8% accuracy. This further demonstrates the added value of fluorescence labeling for revealing common taxonomic traits. The method also provides a confidence score that can be used to filter only high-confidence predictions for in-depth analysis, or to focus on the weakest predictions to generate additional curation efforts (*Figure 3—figure supplement 2*). To further evaluate the value of the features, we computed the classification accuracy using a reduced number of features. We observed that at least 400 features are needed to push accuracy above 82% (see *Figure 3—figure supplement 4*).

Beyond automated assessment of abundance and diversity of micro-eukaryotes across large geographic and taxonomic scales (*Figure 3b*, *Figure 3—figure supplement 3*, *Figure 3—source datas 3–4*), e-HCFM images and descriptors can be used to link fundamental cell biology features to both taxonomy and ecology. For example, whereas organismal biomasses are usually extrapolated from 2D fingerprints for investigating their relationship to nutrient uptake or to carbon flux, our method enables to use 3D biovolumes and/or sub-cellular structures which can be stratified by taxonomy (*Figure 3c*). We further show how concentration of a few major taxa increases differentially with the concentration of phosphate (*Figure 3d*, *Figure 3—source datas 5–7*). These results add global support to previous reports that purely photosynthetic organisms (e.g., Bacillaryophyta) are more dependent on exogenous phosphate than mixotrophic or heterotrophic (e.g., Holodinophyta) populations (*Lin et al., 2016*). A further potential of e-HCFM that complements DNA sequence-based inferences is the ability to directly detect and quantify biotic interactions (*Figure 2*), such as the finding of an unknown nanoflagellate attached to the diatom *Chaeoteros simplex* (*Figure 2—figure supplement 1*).

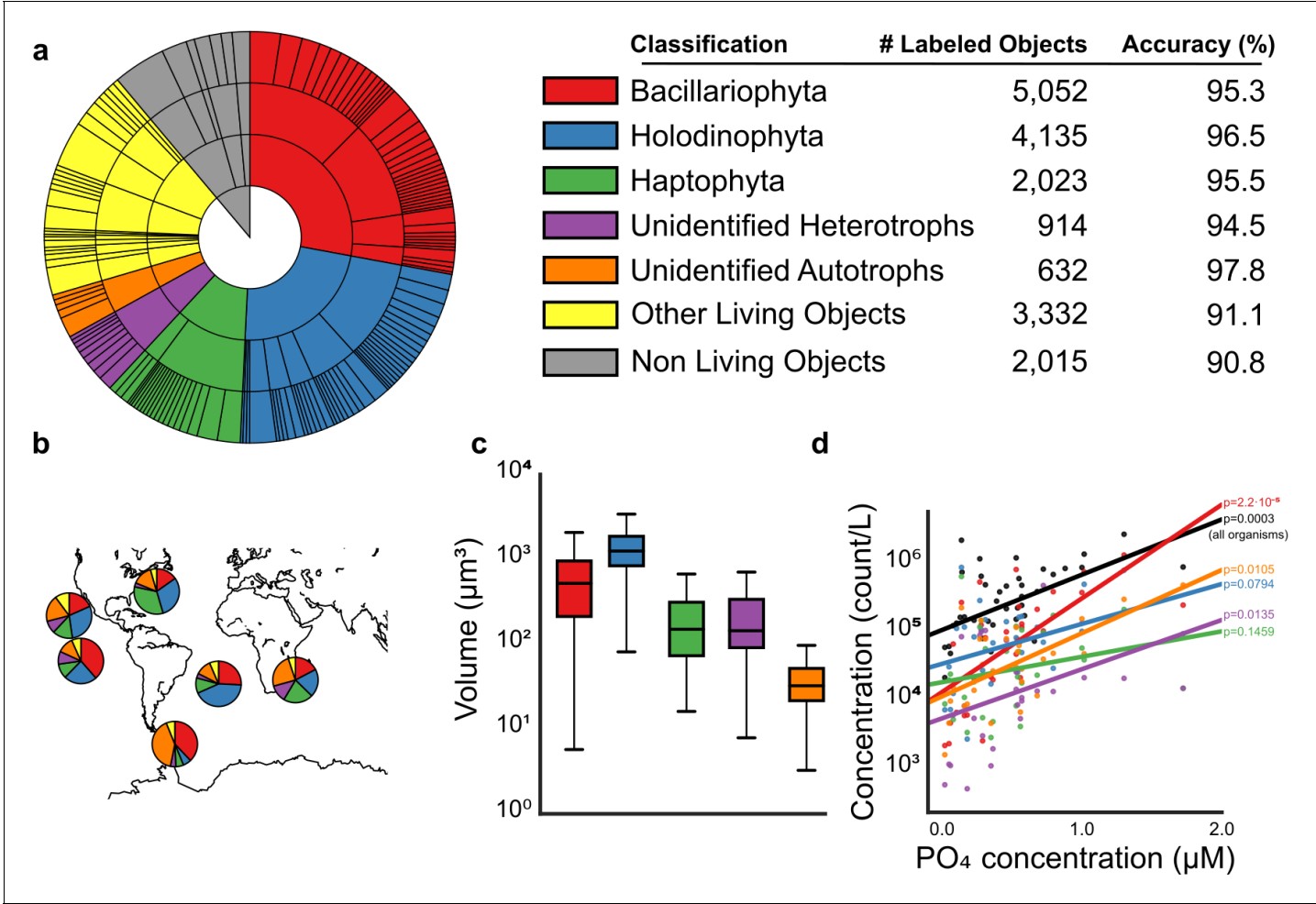

**Figure 3.** Analysis of e-HCFM images and their descriptors. (**a**) Overview of the training set as a hierarchical pie chart. The size of the slices scales with the number of elements in the training set (details in *Figure 3—source data 1*). Accuracy values (%) show the recall for each of the 5-high level groupings considered (see also *Figure 3—figure supplements 1*, *2* and *4* and *Figure 3—source data 1* and *2*). (**b**) Relative abundances of recovered 'live' cells (cells with preserved organellar content, see *Figure 1—figure supplement 5*), per high-level group and per ocean basin (see also *Figure 3—figure supplement 3*, and *Figure 3—source data 3* and *4*). (**c**) Distribution of total cell biovolumes stratified per major taxonomic group. (**d**) Relationship between phosphate concentration and plankton concentration (in counts per liter); black line: total concentration of living organisms in the sample; colored line: total concentration in that taxonomic group (displayed p-values are Spearman correlations, lines are least-squares best-fits, see Methods and *Figure 3—source data 5–7*). [For reviewers - list of information supplements and legends]. (Tables cannot be included in the manuscript and they are provided as separated files).

DOI: https://doi.org/10.7554/eLife.26066.019

The following source data and figure supplements are available for figure 3:

**Source data 1.** Organization of the hierarchical classification scheme for the automated classification, the training set categories abundance and the recall value for each category of the four levels (four tables).
DOI: https://doi.org/10.7554/eLife.26066.028

**Source data 2.** Confusion matrix generated by the classifier at the classification level 4.
DOI: https://doi.org/10.7554/eLife.26066.029

**Source data 3.** Relative abundance of each taxon in each sample.
DOI: https://doi.org/10.7554/eLife.26066.030

**Source data 4.** Assignment of stations to oceanic provinces.
DOI: https://doi.org/10.7554/eLife.26066.031

**Source data 5.** Object counts (normalized to seawater volume) per taxonomic group (panel d).
DOI: https://doi.org/10.7554/eLife.26066.032

**Source data 6.** Measured $PO_4$ concentrations (panel d).
DOI: https://doi.org/10.7554/eLife.26066.033

*Figure 3 continued on next page*

*Figure 3 continued*

**Source data 7.** Values of N, Spearman correlation (rho), and number of samples (N) for each sub-group (panel d).
DOI: https://doi.org/10.7554/eLife.26066.034
**Figure supplement 1.** Binary confusion matrix showing how classification errors are typically within the same broad taxonomical group.
DOI: https://doi.org/10.7554/eLife.26066.020
**Figure supplement 2.** Tradeoff between accuracy and recall.
DOI: https://doi.org/10.7554/eLife.26066.021
**Figure supplement 2—source data 1.** Classification results for all objects in the training data at the fourth (finest) resolution level presented in in left panel (obtained by cross-validation).
DOI: https://doi.org/10.7554/eLife.26066.022
**Figure supplement 2—source data 2.** Classification results for all objects in the training data at the third resolution level presented in right panel (obtained by cross-validation).
DOI: https://doi.org/10.7554/eLife.26066.023
**Figure supplement 3.** Ordination of eHCFM-derived taxonomic abundances reveals regional clustering and tests technical reproducibility of eHCFM-taxonomic profiling.
DOI: https://doi.org/10.7554/eLife.26066.024
**Figure supplement 3—source data 1.** Derived principal component values (original data is *Figure 3—source data 2*).
DOI: https://doi.org/10.7554/eLife.26066.025
**Figure supplement 4.** Classification accuracy as a function of the number of features.
DOI: https://doi.org/10.7554/eLife.26066.026
**Figure supplement 4—source data 1.** Accuracy of classification (estimated by cross validation) using a limited number of features (from 5 to 480, in increments of 5).
DOI: https://doi.org/10.7554/eLife.26066.027

## Conclusion

The main advantage of e-HCFM is the use of 3D fluorescence microscopy in an automated and quantitative manner for environmental microbiology. The relative slowness of the CSLM acquisition can be overcome by widefield microscopy with deconvolution or spinning-disk scanning microscopy, which can both benefit from the high sensitivity and dynamical range of recent cameras. Overall, e-HCFM can be applied to samples from any aquatic environment and be adapted for other habitats. For the sake of taxonomic comprehensiveness we have used a limited set of fluorescent markers highlighting the broad features of eukaryotic cells. However, many other subcellular structures or expressed genes (FISH) can potentially be visualized and quantified by e-HCFM, to provide information about their variation and evolution across the diversity of understudied microbial eukaryotic clades (*del Campo et al., 2014*; *de Vargas et al., 2015*), thus filling the gap between cell biology, evolution, and ecosystem structure and function. A key future challenge will be to combine automated, 3D multicolor fluorescent imaging of complex natural assemblages of eukaryotic cells with online detection of cells of interest and their isolation for single-cell –omics (*Kolisko et al., 2014*). This will allow bridging the deluge of novel environmental eukaryotic genetic data to the complex cellular structure, shape, and behaviour of eukaryotic biota, while scaling up the information, through global meta-omics datasets, to the level of the ecosystems.

## Materials and methods

### Eukaryotic cells sampling and preservation

The environmental samples used in this study were collected during the *Tara* Oceans expeditions (*Pesant et al., 2015*; *Tara Oceans Consortium, Coordinators and Tara Oceans Expedition, Participants, 2017*). At each station, an identical sampling protocol was used to collect and preserve organisms in the 5 to 20 µm size range from surface (5 m depth) and/or deep chlorophyll maximum (DCM) depths. A dedicated net with a 5 µm mesh size was gently towed for 10 min to avoid net saturation and preserve plankton morphology. A flowmeter at net's mouth allowed quantification of the volume of filtered marine water. Concentrated plankton sample in the cod-end was then carefully recovered and poured through a 20 µm sieve to remove larger organisms. The filtrate was resuspended with 0.2 µm-filtered seawater up to 3 L, providing enough material for the various *Tara*

Oceans morpho-genetic protocols (*Pesant et al., 2015*). For e-HCFM, 45 ml were poured into a 50 ml tube pre-aliquoted with 5 ml of 10% monomeric formaldehyde (1% final concentration) buffered at pH7.5 (prepared from paraformaldehyde powder) and 500 µl of 25% EM grade glutaraldehyde (0.25% final concentration) (*Marie et al., 1999*). After a gentle homogenization, the samples were kept at 4°C. This combination of formaldehyde and glutaraldehyde balances fixative penetration and cell stiffening (*Kiernan, 2000*). Live imaging was based on both cultures of marine protist from the Roscoff Culture Collection (RCC) and specimen isolated from the environment (Villefranche-sur-mer, France).

## Cells' mounting and fluorescent labeling

The labeling/mounting protocol was optimized to minimize operational steps and cell manipulations. After a gentle resuspension of the cells in the sample tube, an aliquot was loaded into a 8-well Lab-Tek II chambered cover glass (Nunc 155382; Thermo Fisher Scientific, MA, USA), with bottom pre-coated with Poly-L-lysine (Sigma-Aldricht P4707; Merck, Germany). This optical multi-well plate allows loading about 1 ml of sample per cm$^2$. In particular, two *Tara* Oceans samples were placed in separated wells of the same plate. Four other wells were dedicated to fluorescent beads as internal calibration standards of fluorescent signals (see below). A gentle centrifugation (1,000 rpm) of the plate on a swinging bucket rotor allowed settling down of planktonic particles onto the sticky poly-L-lysine layer. This concentrated the objects on the well bottom while decreasing their orientation variability in relation to the optical axis (*Figure 1—figure supplement 3a*). The volume of sample in each well was empirically adjusted by visual inspection of the wells for avoiding cell saturation (*Figure 1—figure supplement 3c*).

The sample was then washed for 5 min with 1 ml of artificial seawater (AS media recipe (*Berges et al., 2001*) without vitamins and trace metals) to remove the fixative. All washing steps were performed using pipette tips capped by a 1 µm mesh piece to avoid cell losses. AS washing maintains medium buffering capacity and an ionic composition close to natural seawater (without any organic material), thus preventing modifications of cell shape and dissolution of mineral covering. The cells were first stained with 0.1 mg.ml$^{-1}$ of a Poly-L-lysine conjugated with Alexa Fluor 546 (see below) in 500 µl of AS for 15 min, and then washed with 1 ml of AS for 5 min. Further fluorescent labeling was performed, for at least 30 min, with 500 µl of an AS solution containing 10 µM of Hoechst33342 (Invitrogen H21492; Thermo Fisher Scientific, MA, USA) and 1.5 µM of DiOC6(3) (Invitrogen D273; Thermo Fisher Scientific, MA, USA). These dyes do not require washing, reducing again sample manipulations.

## A universal fluorescent dye for eukaryotic cell cover

We developed a fluorescent labeling compound able to delineate the fine structures of all protist surfaces independently of their biochemistry. The poly-L-lysine (PLL) conjugated with Alexa Fluor 546 (see below) was found to be an effective probe for this task. α-poly-L-lysine is a cationic polymer under neutral pH conditions, but the length of the lysine aliphatic side chain confers a slight amphiphilic behavior to this polymer. Electrostatic binding occurs thus with both mineral (e.g. silica, calcium carbonate, strontium sulfate) and organic (e.g. DNA, proteins, or polysaccharides) materials. Furthermore, the glutaraldehyde used to fix the plankton sample contributes to crosslink PLL to cellular proteins. Examples of the labeling efficiency of Alexa-PLL are shown in *Figures 1* and *2* and *Figure 1—figure supplements 1*, *2* and *5* and *Figure 2—figure supplement 1*. Note that a broad choice of dye are available for protein conjugation (e.g. the Alexa Fluor family), allowing selection of spectrally relevant fluorochromes, a convenient feature for combination with other dyes in multi-channel methods.

### Conjugation protocol

The fluorescent dye moiety is an Alexa Fluor 546 succinimidyl ester (AF546SE, Invitrogen A20002; Thermo Fisher Scientific, MA, USA), providing an efficient way to selectively link the Alexa Fluor dyes to primary amines (R-NH2) located on the lysine side chain. The PLL (Sigma-Aldrich P5899; Merck, Germany) polymer was selected for its high size range (>300,000 mol wt/2,340 lysine unit), limiting penetration of the dye into the cell. The conjugation reaction was performed according to the AF546 provider's protocol for protein conjugation. Briefly, AF546SE was prepared in dry

DMSO at 10 mM, and 5 µl of this solution was mixed with 1 ml of a 10 mg.ml$^{-1}$ PLL solution (in NaHCO3 solution at 0.1M pH8.3) for 1 hr at 20°C under gentle shaking. The ratio [R-NH2]/[AF546SE] was calculated to generate the statistical binding of one AF546 per 1500 lysine units. Such a ratio, combined with the average size of PLL (>2340 lysine units) should generate a minimal labeling rate of 1 AF546 per PLL molecule. The stock solution of this conjugate was kept at −20°C.

## Automated 3D fluorescence microscopy

High throughput imaging leads to a trade-off between time constraints, the amount of region of interest, and the precision of information that can be extracted for each of them. In this study, we optimized settings for an increased throughput rather than spatial resolution. Microscopy was conducted using a commercially available inverted SP8 laser scanning confocal microscope (Leica Microsystem, Germany) equipped with a compact supply unit which integrates a LIAchroic scan head and several laser lines (405 nm, 488 nm, 552 nm, 638 nm). A two-step sequential acquisition was designed to collect the signal from five channels. The first step aims at recording DiOC6(3) signal (Ex488 nm/Em505-520 nm) simultaneously with the chlorophyll autofluorescence signal (Ex638 nm/ Em680-720 nm), and the transmitted light channel. The second step is then dedicated to acquisition of the Hoechst signal (Ex405 nm/Em415-475 nm) and the AF546 signal (Ex552 nm/Em570-590 nm). AF546 is poorly excited by 488 nm and 638 nm beam lines and its emission was not detected between 505 and 520 nm, and between 680 and 720 nm. The chlorophyll fluorescence was recorded in the spectral range 680–720 nm where AF546 and Hoechst do not emit with 488 nm and 638 nm illumination. Chlorophyll is also excited at 405 nm but it does not emit in both the hoechst (415–475 nm) and AF456 (570–590 nm) channels. The hoechst fluorescence spectrum is broad and may slightly bleed through the AF546 channel. However the brightness of AF546 signal imposed low sensitivity settings which reduced the impact of other dim signals that might bleed through this channel. The potential signal from Hoechst was finally not subtracted because the AF546 fluorescence was mostly used to delineate cell edges.

The laser power and exposure settings were tuned to reach a tradeoff between the dimmest and brightest object of interest whereas the limited dynamic range of our photomultiplier tube detector (PMT) cannot avoid some signal saturation (the variability of environmental samples cannot be anticipated). Signals were not averaged. The automation was piloted through the HCS A module of the LASAF software (Leica Microsystem). We used the water immersion lens HC PL APO 40x/1,10 mot CORR CS2. The scanning was bidirectional with a speed set at 600 Hz. The pinhole was adjusted to 1 Airy unit for all channels. Each square field of view (fov) is 385.62 µm wide and 21.80 µm thick. Field of views were organized in a rectangular mosaic with 10% overlapped in X and Y axes. The spatial sampling frequency was 0.188*0.188*1.090 µm voxel size (2048*2048*per frame). The Z-stack steps match the full width half maximum as a measure of the optical slice thickness at our lower signal emission wavelength (415 nm, refractive index 1.33, numerical aperture 1.1, 1 Airy unit) while the pixel size matches the objective XY resolution. This is larger than the Niquist sampling rate (75*75*300 nm, with $\lambda_{emission}$ at 415 nm) to maximize throughput. A software autofocus procedure estimated the interface plane separating the sample from the coverslip. Based on this estimate, we initiated the Z-stack at a −1 µm offset.

### Monitoring of the microscope performance

Since we processed the *Tara* Oceans samples sequentially in multiple runs, we monitored the stability of the illumination (laser sources), optics and the PMTs by using fluorescent beads (Invitrogen I7225, InSpeck deep-red, lot1267301; Thermo Fisher Scientific, MA, USA) as internal calibration standards which were included in each multi-well plate prior to running image acquisition (see mounting details above). We show in the *Figure 1—figure supplement 6* (see also *Figure 1—figure supplement 6—source data 1*) the distribution of fluorescence intensity for 3 types of beads (0.32%; 1.41%; 4.10% fluorophore concentration) emitting in the chlorophyll channel (average intensity after sum projection of corresponding z-stacks, segmentation and size filtering). Further normalization of the chlorophyll fluorescence quantification were not required for comparing samples as the intensities measured looked very stable.

## Automated image analysis

At a high-level, image analysis proceeds in three steps: (1) identification of objects, (2) computation of per-object features, and (3) classification of objects.

### Identification of objects

First, an estimate of background intensity is computed. For each fluorescence channel, we compute the average voxel value and the standard deviation of voxel intensities. A per-channel threshold is then defined as the mean plus one-and-a-half standard deviation (or the intensity value 1, if the result of the computation is below 1). This method was chosen based on previous work which identified the mean as a good threshold for fluorescence microscopy images (*Coelho et al., 2009*). Tests with Otsu thresholding led to fewer objects being detected (231,610 compared to 336,655 with the chosen method). Objects are then identified in a two-dimensional projection of the image. For this projection, we use a robust variation of the traditional max-projection method (which we called the almost-max projection). Namely, from a three dimensional image I(x,y,z), we build a two-dimensional project P(x,y) by taking all the pixels I(x,y,z), sorting them and taking the second highest value. The vignettes used for display are also generated using the almost-max projection. This projection is then median filtered to reduce noise and thresholded using the previously computed threshold to obtain a binary masks for each channel.

Binary masks from all channels are combined (using the OR operation), morphologically filtered to reduce noise and labeled to identify objects (an open and a close operation are sequentially performed with a 4-connected structuring element; followed by morphologically closing holes). These objects are then size filtered to remove objects smaller in area than particles of interest (here 12.6 $\mu m^2$ as this corresponds to a circle of 5 $\mu m$ diameter). Objects touching any of the four image borders are also removed. Finally, because the imaging was performed in overlapping fields, it is possible to detect the same object in multiple fields, thus we assign it to its canonical field based on the geometry and remove it from all other fields (see *Figure 1—figure supplement 4a*). Image analysis was performed using the mahotas package (*Coelho, 2013*).

### Computation of per object features

Each object is processed independently to extract sub-objects and features. Sub-objects are identified using the same thresholding procedure as above (however, this is now applied on the three dimensional volume after median filtering). Sub-objects are filtered to remove any object whose volume is smaller than 0.5 $\mu m^3$ (circa the volume of a sphere of radius 0.5 $\mu m$). Features are computed on both the projected images (computing Haralick textures, linear binary patterns, Zernike moments, and morphological features) and on the three dimensional volumes (for the estimation of bio-volumes and overlap features). See *Figure 1—source data 2* for a complete list of features.

### Object classification

A hand-labeled dataset of 18,103 objects (5.4% of the whole dataset) was used for training an automated classifier. For maximizing the diversity of the training set, the entire dataset was explored, similar objects were categorized and 155 categories with more than 30 specimens were kept. We did not overweight classes of very abundant object and considered that few hundreds of representative objects were enough to define a class statistically. These categories were organized into a four-level hierarchical framework (*Figure 3a*; *Figure 3—source data 1*) reflecting taxonomic lineage for organisms and particle types for other objects. The goal of this hierarchical pattern of classification was to evaluate the classifier performance at different granularities. A random-forest classifier, using 500 trees (*Breiman, 2001*) was used for classification. This classification method was chosen empirically as the best combination of accuracy and computational performance among several methods tested (support-vector machine with diverse set of kernels, logistic regression, and random forests). This step of the pipeline was implemented using the scikit-learn package (*Pedregosa, 2011*). At classification time, the software returns, for each object, a probability estimate for each class. The object is assigned to the class with the highest probability; a confidence score is defined as the difference between the highest and the second-highest probability values. Feature importance was computed by the scikit-learn package as the Gini importance (*Breiman, 2001*). We report a normalized version where the reported importance values were divided by the average importance so that

values above one correspond to features with higher than average importance and values below one to features less important than average (*Figure 1—source data 2*). To test the effect of reducing the number of features, we used cross-validation, computing the feature importance independently in each cross-validation fold.

## Computational costs and distributed computation

In total, it takes circa 40 CPU hours to process one acquisition (which consists of two wells, 768 data fovs, and 48 control fovs). However, after the background estimation step, the process is embarrassingly parallel as each fov can be processed in parallel (a computationally inexpensive post-processing of the data relabels all objects so that they are numbered sequentially within each sample). Our software is designed to take advantage of a multi-core machine computer cluster by automatically splitting the computation across available CPUs. Image analyses of one microscopy run are performed in 20 min with sufficient resources.

## Statistical analysis of relationship between phosphate concentration and organism counts

All organisms from a single sample were considered (data points from multiple technical replicates of the same sample, when available, were concatenated). Counts were normalized by the volume of seawater represented by the imaged sample(s). Correlations were evaluated using Spearman correlation (implemented by Python's scipy.stats module). Samples which do not contain the relevant taxonomic group were excluded from each taxonomic-specific analysis.

## Accession URLs for software and data sharing purposes

The software code is made available under an open source license (https://git.embl.de/coelho/eHCFM). All raw images and preprocessed thumbnails are publicly available at EBI-BioStudies (accession ID: S-BSST51, https://www.ebi.ac.uk/biostudies/studies/S-BSST51). Images can also be explored through the Image Data Resource platform (*Williams et al., 2017*; https://doi.org/10.17867/10000108). The preprocessed thumbnails will be released for online expert annotations at EcoTaxa (http://ecotaxa.sb-roscoff.fr), and taxonomically and ecologically informed thumbnails will be available for public exploration. The inventory of experiment metadata, samples and their associated contextual data from the Tara Oceans expedition are available at PANGAEA (*Tara Oceans Consortium, Coordinators and Tara Oceans Expedition, Participants, 2017*; https://doi.org/10.1594/PANGAEA.881193). The eHCFM dataset presented herein constitutes an additional Tara Oceans global resource available for the wider community, and complementing other eco-morphogenetic datasets generated in the project toward a holistic understanding of the world plankton (*Pesant et al., 2015*).

## Acknowledgements

We thank the following sponsors for their commitment: the French Ministry of Research and Government through the 'Investissements d'Avenir' program OCEANOMICS (ANR-11-BTBR-0008); The ANR program POSEIDON (ANR-09-BLAN-0348), CNRS (in particular, the GDR3280); EMBL; UPMC; and VEOLIA for providing seed-funding at early stages of this project. We thank L Santangeli and A Tanaka for sample preparations and help, V Hilsenstein, C Conrad, and S Terjung from the EMBL Advanced Light Microscopy Facility for precious assistance at the microscope, Yan Yuan for IT support, EMBL IT facility and Christian Boulin for providing secured data storage. This article is contribution number 60 of *Tara* Oceans.

## Additional information

### Funding

| Funder | Grant reference number | Author |
|---|---|---|
| VEOLIA Foundation | | Sebastien Colin |

| | | |
|---|---|---|
| Centre National de la Recherche Scientifique | | Sebastien Colin<br>Chris Bowler<br>Colomban de Vargas |
| Agence Nationale de la Recherche | ANR-11-BTBR-0008 | Colomban de Vargas<br>Sebastien Colin<br>Chris Bowler<br>Shinichi Sunagawa<br>Peer Bork<br>Eric Karsenti |
| European Molecular Biology Laboratory | | Luis Pedro Coelho<br>Shinichi Sunagawa<br>Eric Karsenti<br>Peer Bork<br>Rainer Pepperkok |
| Université Pierre et Marie Curie | | Colomban de Vargas |
| Agence Nationale de la Recherche | ANR-09-BLAN-0348 | Colomban de Vargas |

The funders had no role in study design, data collection and interpretation, or the decision to submit the work for publication.

## Author contributions
Sebastien Colin, Conceptualization, Methodology, Investigation, Data curation, Formal analysis, Software, Visualization, Funding acquisition, Writing—original draft, Writing—review and editing; Luis Pedro Coelho, Methodology, Investigation, Data curation, Formal analysis, Software, Visualization, Writing—original draft, Writing—review and editing; Shinichi Sunagawa, Formal analysis, Visualization, Supervision, Funding acquisition, Writing—review and editing; Chris Bowler, Funding acquisition, Writing—review and editing; Eric Karsenti, Conceptualization, Methodology, Resource, Supervision, Funding acquisition, Writing—review and editing, Project administration; Peer Bork, Resource, Supervision, Funding acquisition, Writing—review and editing; Rainer Pepperkok, Methodology, Resource, Supervision, Writing—review and editing; Colomban de Vargas, Conceptualization, Methodology, Formal analysis, Resource, Supervision, Funding acquisition, Writing—review and editing, Project administration

## Author ORCIDs
Sebastien Colin (ID) https://orcid.org/0000-0003-4440-9396
Luis Pedro Coelho (ID) https://orcid.org/0000-0002-9280-7885
Colomban de Vargas (ID) https://orcid.org/0000-0002-6476-6019

## Decision letter and Author response
Decision letter https://doi.org/10.7554/eLife.26066.042
Author response https://doi.org/10.7554/eLife.26066.043

# Additional files

## Supplementary files
• Transparent reporting form
DOI: https://doi.org/10.7554/eLife.26066.035

## Major datasets
The following datasets were generated:

| Author(s) | Year | Dataset title | Dataset URL | Database, license, and accessibility information |
|---|---|---|---|---|
| Sebastien Colin, Luis Pedro Coelho, Shinichi Sunagawa, Chris Bowler, Eric Karsenti, Peer Bork, Rainer Pepperkok, Colomban de Vargas | 2017 | idr0015-colin-taraoceans/screenA | https://idr.openmicroscopy.org/webclient/?show=screen-1201 | Publicly available at Image Data Resource platform (doi: 10.17867/10000108) |
| Colin S, Coelho LP, Sunagawa S, Bowler C, Karsenti E, Bork P, Pepperkok R, de Vargas C | 2016 | eHCFM–TARA_OCEANS–HCS1–H5 | https://www.ebi.ac.uk/biostudies/studies/S-BSST51 | Publicly available at EBI-BioStudies (accession no. S-BSST51) |

The following previously published dataset was used:

| Author(s) | Year | Dataset title | Dataset URL | Database, license, and accessibility information |
|---|---|---|---|---|
| Tara Oceans Consortium Coordinators;Tara Oceans Expedition Participants | 2016 | Registry of all samples from the Tara Oceans Expedition (2009-2013) | https://doi.pangaea.de/10.1594/PANGAEA.859953 | Publicly available at PANGAEA (doi:10.1594/PANGAEA.859953). |

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
