## [Decision Letter]

Thank you for submitting your article "Quantitative 3D-Imaging for Cell Biology and Ecology of Environmental Microbial Eukaryotes" for consideration by *eLife*. Your article has been reviewed by two peer reviewers, and the evaluation has been overseen by a Reviewing Editor and Wendy Garrett as the Senior Editor. The following individual involved in review of your submission has agreed to reveal her identity: Heidi Sosik (Reviewer #1).

The reviewers have discussed the reviews with one another and the Reviewing Editor has drafted this decision to help you prepare a revised submission.

The reviewers made a number of constructive comments, which you should carefully address. These include:

Essential Revisions:

1) At the review stage, it was extremely difficult to find, understand, and evaluate the supplementary tables and videos. In place of the links on the download page, the editorial office sent me a zip file of the tables and videos with filenames that match the designations in the main manuscript and in your supplementary information, but some of those appear to be mislabeled or to contain the wrong information. For example, the file "Figure 3—source data 1" looks like it just contains the statistical values listed for Figure 3—figure supplement 3—source data 1; whereas that latter file contains the principal component values from the figure. These all need to be checked and their contents appropriately described.

2) Samples were fixed in a high concentration of glutaradehyde (25%) which contributes strong autofluorescence, especially in the green region of the spectrum, that may confound the specific (dye-based) fluorescent signals. Have the authors compared the current images to images of samples fixed only with paraformaldehyde? Please comment.

3) The authors extract a large number of features from each sample. Have they performed a simple principal component analysis to determine which features are most informative? I would suspect that most of the discrimination power is contained within a relatively small subset of these parameters. Please discuss.

4) The thresholding procedure ("mean + 1.5 standard deviations") is arbitrary and depends on the density of fluorescent objects in the image as well as the level of the background. How is the fidelity of the classification impacted by the threshold level? Classification results using at least one other threshold level (e.g. mean+3 standard deviations) or an automated thresholding algorithm should be presented and discussed.

5) The authors claim to have imaged the "entire extent" of each well. However, the bright-field images will be strongly degraded near the edges of the wells due to the cone of transmitted light hitting the vertical edges of the chambers and thus missing the condenser. How was this issue addressed? If this is a simple case of over-statement, then a more accurate description should be used.

6) In terms of throughput, did the authors assess the utility of simple widefield microscopy (with deconvolution) as an alternative to confocal? This will allow for much faster data acquisition. This warrants some discussion.

7) Did the authors correct for channel bleed-through in the 4-color images? This is especially the case when using DiO and AF546. This combination may introduce some cross talk, confounding the classifier.

8) Did the authors compare the static threshold of 1.5 standard deviation above average with a more robust/sophisticated algorithm such as Otsu's method?

9) Very often, classifiers are built such that 50% of the data set is used to train, while the remaining 50% is used to test. Was there a reason the training set was such a small portion of the total data? Was this limited by manually classifying the organisms?

In summary, the reviewers generally appreciated this paper and we look forward to receiving a revised manuscript.

---

## [Author Response]

Essential Revisions:1) At the review stage, it was extremely difficult to find, understand, and evaluate the supplementary tables and videos. In place of the links on the download page, the editorial office sent me a zip file of the tables and videos with filenames that match the designations in the main manuscript and in your supplementary information, but some of those appear to be mislabeled or to contain the wrong information. For example, the file "Figure 3—source data 1" looks like it just contains the statistical values listed for Figure 3—figure supplement 3—source data 1; whereas that latter file contains the principal component values from the figure. These all need to be checked and their contents appropriately described.

We have cross-checked all caption and files. Reviewer was correct that we had incorrectly labeled the data files corresponding to “Figure 3—figure supplement 3”. We have now both corrected the labeling and added the full relative abundance matrix. We have also two additional tables with the averaged measurements depicted in the panel 3D of Figure 3. Note that the full raw data (including the assignment of each object to a taxonomic group and the raw feature measurements) is, as before, provided through *EcoTaxa* due to its large size.

2) Samples were fixed in a high concentration of glutaradehyde (25%) which contributes strong autofluorescence, especially in the green region of the spectrum, that may confound the specific (dye-based) fluorescent signals. Have the authors compared the current images to images of samples fixed only with paraformaldehyde? Please comment.

Reviewer is right in mentioning the green/yellow emitted fluorescence induced by the formaldehyde and glutaraldehyde fixative which occurs mainly under blue and green excitation [see Lee, K., Choi, S., Yang, C., Wu, H. C., and Yu, J. Autofluorescence generation and elimination: a lesson from glutaraldehyde. Chemical Communications, 49(18), 3028-3030 (2013)]. However, we did not use a final concentration of 25% Glutaraldehyde, but a mix of 1% formaldehyde (prepared from paraformaldehyde) and only 0.25% Glutaraldehyde, as mentioned in the first version. We have modified the sentence to avoid confusion between stock concentration and working concentration: “For e-HCFM, 45 ml were poured into a 50 ml tube pre-aliquoted with 5 ml of 10% monomeric formaldehyde (1% final concentration) buffered at pH7.5 (prepared from paraformaldehyde powder) and 500 μl of 25% EM grade glutaraldehyde (0.25% final concentration) [Maria et al., 1999].”

We did not compare our low-glutaraldehyde protocol to a protocol using paraformaldehyde only, as it is anyways very hard to disentangle signals from native autofluorescences (e.g. green autofluorescence or fluorescence from photosynthetic pigments like phycobiliproteins), which can be strong and out of prediction/control in environmental samples. However, we strongly decreased the impact of glutaraldehyde autofluorescence by minimizing its final concentration. In addition, both the DiOC6 and AlexaFluor546 staining overlap the glutaraldehyde excitation/emission spectra, but they generate much brighter fluorescence signal than the glutaraldehyde autofluorescence, further reducing its overall contribution. Note that the signal quantification of these two channels was mostly used for classification rather than function quantification.

In order to quench the autofluorescence from glutaraldehyde (coming from double bond with primary amines and free aldehyde groups), the fixative could also be reduced by a NaBH4 treatment. However, this treatment would negatively impact other critical parts of our protocol: the generation of H_2_ gas bubble can interfere with cell attachment to the bottom of the well; preserving free aldehyde groups contributes to cross-link poly-L-lysine with cellular proteins.

3) The authors extract a large number of features from each sample. Have they performed a simple principal component analysis to determine which features are most informative? I would suspect that most of the discrimination power is contained within a relatively small subset of these parameters. Please discuss.

We thank the reviewer for this question as it prompted us to quantitatively address it. We used the random forest method to select features that are most discriminative and to test increasing number of features (selected from most informative to least informative). The results demonstrate that almost 200 feature are necessary until accuracy similar to the full dataset is reached and that even after 400 features there is still some improvement observed.

Note too that features are computed in groups and features from the same group may have differing discriminatory power. We find overlap features (features computed from the overlap of fluorescence between channels) to be present both among the most discriminative and least discriminative features. Similarly, the zero-th Zernike moments (effectively, average fluorescence after normalization) are not very discriminative, but other Zernike moments are much more important.

We have now summarized this analysis in the main text (Results and Discussion, fourth paragraph) and added Figure 3—figure supplement 4 as a supplemental figure (with its corresponding source data, Figure 3—figure supplement 4—source data 1).

4) The thresholding procedure ("mean + 1.5 standard deviations") is arbitrary and depends on the density of fluorescent objects in the image as well as the level of the background. How is the fidelity of the classification impacted by the threshold level? Classification results using at least one other threshold level (e.g. mean+3 standard deviations) or an automated thresholding algorithm should be presented and discussed.

We chose this procedure based on previous work in fluorescence image analysis which showed that the mean could outperform methods based on automated thresholding algorithms such as Otsu [Coelho, Aabid and Murphy, 2009]. This value is computed for the whole plate and reused for the different fields since we did not expect (and did not observe) any large differences in background fluorescence between fields. Since background occupies the majority of the volume, this can provide a good separation between background and objects. Due to the high variance in object brightness, we feared that a method such as Otsu may erroneously classify dim objects as background. Indeed, quantitatively, using Otsu results in much fewer objects being detected (231,610 compared to 336,655 with the method in the text).

We followed the reviewer’s suggestion and implemented the threshold using the “mean + 3 standard deviations” and the segmentation is similar enough that the classification system achieves almost identical accuracy (82.1% compared to 82.2% for the version in the text). When using Otsu, we see a slight decrease (78.5%), which may be due to the fact that several objects in the training data are no longer detected (see above).

In the revised version, we have added the following two sentences justifying our choices 1) “This method was chosen based on previous work which identified the mean as a good threshold for fluorescence microscopy images [Coelho, Aabid and Murphy, 2009]; 2) “Tests with Otsu thresholding led to fewer objects being detected (231,610 compared to 336,655 with the chosen method).”

5) The authors claim to have imaged the "entire extent" of each well. However, the bright-field images will be strongly degraded near the edges of the wells due to the cone of transmitted light hitting the vertical edges of the chambers and thus missing the condenser. How was this issue addressed? If this is a simple case of over-statement, then a more accurate description should be used.

We confirm that the signal from transmitted light is degraded for the field of view in the vicinity of the well edge. The meniscus of the liquid in the well also impacts this signal depending on the position of the objective. Nonetheless, in terms of classification, the information extracted from the bright field channel is still valuable (perhaps because many of the features are robust to illumination changes). As we describe in the text (Results and Discussion, fourth paragraph), we obtain 56.8% accuracy using only the bright field images. Even though this is much worse than the results when combining all channels (including the fluorescence channels), it is still significantly better than random (recall that we have 155 classes). To further explore this question, we attempted classification while excluding the features from the bright field channel. The result was slightly worse accuracy (82.0% vs. 82.2%).

6) In terms of throughput, did the authors assess the utility of simple widefield microscopy (with deconvolution) as an alternative to confocal? This will allow for much faster data acquisition. This warrants some discussion.

The reviewer is right. Acquisition by widefield microscopy with deconvolution was considered when we initiated the project a few years ago. Unfortunately, the 3D deconvolution of large fields of view from fluorescence microscopy datasets required intensive computation (several seconds per stacks and per channels). The time saved at the acquisition level was lost at the computation level without having the optical sectioning performance and the excitation/emission versatility of our SP8 CLSM setup. We also explored the use of spinning disk systems, equipped with a Yokogawa CSU-X1 Confocal Scanner Unit. The time saved in scanning at similar resolution was lost in the multiplication of overlapping positions because the fields of view were much smaller than the one acquired with the SP8 CLSM setup.

However we agree that both widefield and spinning disk microscopy deserve further evaluation in the future. Recent GPU-based deconvolution algorithms [see Bruce, M. A. and Butte, M. J. Real-time GPU-based 3D Deconvolution. Opt. Express 21, 4766–4773 (2013).] and open source softwares [see Sage, D. et al. DeconvolutionLab2: An open-source software for deconvolution microscopy. Methods 115, 28–41 (2017).] are making the widefield microscopy approach more time-efficient and economically attractive; both methods can also take advantage of the higher sensitivity and dynamical range of the latest camera as compared to the photomultiplier tubes used in CSLM. Following the reviewer’s suggestion, we now mention this in the Conclusion.

Note that we are investigating another way to speed up the acquisition process by implementing a 2-steps acquisitions in e-HCFM. The first step is a fast prescan of the cell preparation aiming at spotting the object of interest based on a reduced set of features. The second step triggers dedicated imaging job at positions of interest. This method significantly improves the triangular trade off: ‘imaging time / information quality / quantity of object’, while still taking advantage of the great optical sectioning performance and the ex/em versatility of CLSM. It is specifically adapted for rare events or discrete distribution of cells without wasting time in imaging at high resolution empty area. It also significantly decreases image storage requirement and associated cost. We are not mentioning this work in progress in this paper presenting the primary version of e-HCFM.

7) Did the authors correct for channel bleed-through in the 4-color images? This is especially the case when using DiO and AF546. This combination may introduce some cross talk, confounding the classifier.

In our protocol, the recording of DiOC(6) and AlexaFluor546 fluorescence were separated sequentially through 2 acquisition steps. The first step combines an excitation at 488 nm and 638 nm which allows the measurement of DiO emission signal between 505-520 nm and chlorophyll autofluorescence between 680-720 nm. We now mention: “AF546 is poorly excited by 488 nm and 638 nm beamlines and its emission was not detected between 505-520 nm and between 680-720 nm. The chlorophyll fluorescence was recorded in the spectral range 680-720 nm where AF546 and Hoechst do not emit with 488 nm and 638 nm illumination.”

However a low autofluorescent noise can be revealed by maximum projection in this Chlorophyll channel (see Figure 1—figure supplement 5) which seems related to core cell structures. The origin of this signal is not clear but its detection was caused by the selection of high sensitivity settings to enable the detection of dim chlorophyll signal. Similarly, the second step excites simultaneously Hoechst at 405 nm and AF546 at 552 nm. We now mention: “Chlorophyll is also excited at 405 nm but it does not emit in both the hoechst (415-475 nm) and AF456 (570-590 nm) channels. […] The potential signal from Hoechst was finally not subtracted because the AF546 was mostly used to delineate cell edges.”.

We would also note that as long as any cross-talk or auto-fluorescence is consistent within the same taxonomic group, the classification system will be robust.

8) Did the authors compare the static threshold of 1.5 standard deviation above average with a more robust/sophisticated algorithm such as Otsu's method?

See response to point 4.

9) Very often, classifiers are built such that 50% of the data set is used to train, while the remaining 50% is used to test. Was there a reason the training set was such a small portion of the total data? Was this limited by manually classifying the organisms?

While it is true that the training set represents a small fraction of the whole dataset, we consider that this reflects rather the large size of the dataset. In absolute terms, though, we consider 18,103 manually labeled objects as a large training set.

In addition, we carefully screened the whole object collection for optimizing the diversity of classes in the training set, and created classes each time a group of similar objects reached at least 30 (subsection “(3) Object classification”). On the other hand we did not overweight classes of very abundant object and considered that few hundreds of representative objects were enough to define a class statistically which is now mentioned in the aforementioned subsection.